# Health Crisis and the Dual Reflexivity of Knowledge

Denis Bernardeau-Moreau 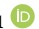

Multidisciplinary Research Unit for Sport, Health, Society (URePSSS), University of Lille, ULR 7369, F-59000 Lille, France; denis.bernardeau-moreau@univ-lille.fr

**Abstract:** Although successive pandemic episodes adversely affect populations and it remains difficult to assess their long-term extent and impact, they may, paradoxically, have a positive effect. In fact, they can promote awareness by reviving a form of reflexivity with respect to public health, economic and social policies, and by driving in-depth reflection on the measures that must be taken to limit the current and future imbalances caused by human activity. Habermas emphasises that the reflexive "push" is no longer just a matter for experts; it is also collective, historical, and political, in the sense that it involves citizens who intend to weigh in on the debate and make their voices and wishes heard by policy-makers and economic actors. Reflecting upon the ethics of responsibility (Weber) is therefore essential. If we are to follow Giddens and Habermas' thinking, this reflexivity represents an integral part of the modern age. Our intention, in this article, is to show how major events, beyond an initial period of shock, can help to awake different levels of reflexivity in individuals.

**Keywords:** health crisis; awareness; dual reflexivity of knowledge



## 1. The Anxiety-Inducing Context of the Health Crisis

The number of scientific publications released increases exponentially during periods of an ongoing pandemic. According to the Scopus database, there were nearly 800 articles at the peak of Severe Acute Respiratory Syndrome (SARS) in 2003 and 980 articles at the peak of Middle East Respiratory Syndrome (MERS) in 2014/2015. In 2020, when COVID-19 cases were multiplying rapidly, the number of publications grew to 2000. According to the World Health Organization (WHO), almost 15,000 publications in total have been published since the start of the coronavirus pandemic. This health crisis, which is being felt on a global scale, has brought about a sharp increase in the stress felt by the population. A number of studies (Cohen et al. 1983; Conley et al. 2013; Eicher et al. 2014) reveal high stress levels among students during their university studies. According to these surveys, 35% of students in higher education say that they are anxious and 30% even admit to being depressed. A survey conducted by the American College Health Association[1] shows that stress has become the most serious barrier to studying among American students. This stress is also felt by the general population. According to a survey commissioned by the American Psychological Association in 2021[2], 32% of adults admit to being stressed by the pandemic. A quarter of those surveyed stated that stress impacts their eating habits and their ability to carry out their life plans. According to a study published in The Lancet in October 2021[3], cases of depression and anxiety increased in 2020 by more than 25% worldwide due to the coronavirus pandemic. The world of work has also been severely affected. Interruptions of work activities due to lockdowns and the increasing frequency, then subsequent normalisation, of remote working and virtual meetings between colleagues have considerably disrupted working habits and professional relationships. According to a survey conducted by Dynamic Workplace and the Speak & Act school[4], people in employment feel twice as stressed as they did before the health crisis. Remote working has, in particular, brought about a further blurring of the lines between people's professional and private lives. Over the course of two years, the percentage of employees who are

contacted outside of their working hours has risen from one quarter to one third. The survey highlights the fact that prior to the COVID-19 crisis, 92% of employees felt that digital tools made their daily lives easier. However, only 60% feel this way today. While the aforementioned studies show that successive lockdowns have generated a great deal of stress and anxiety, there has also been a significant increase worldwide in situations involving loneliness and depression, alcohol consumption, drug use, and cases of self-harm or suicidal behaviour.

## 2. The "Revival" of a Collective Reflexivity

Although successive pandemic episodes adversely affect populations and it remains difficult to assess their long-term extent and impact, they may, paradoxically, have a positive effect. In fact, recent studies (Aykut et al. 2010) show that they also promote awareness by reviving a form of reflexivity with respect to public health, economic and social policies, and by driving in-depth reflection on the measures that must be taken to limit the current and future imbalances caused by human activity. Our intention, in this article, is to show how major events, beyond an initial period of shock, can help to awaken different levels of reflexivity in individuals. Habermas[5] emphasises that the reflexive "push" is no longer just a matter for experts; it is also historical and political in the sense that it involves citizens who intend to weigh in on the debate and make their voices and wishes heard by policy-makers and economic actors. Health issues, which are closely linked to issues concerning the future of the planet and that of humans and animals, are now working their way, as announced Habermas, into our collective consciousness. In sociological terminology, reflexivity has several meanings; external when it concerns the social conditions of the production of sociological knowledge in modern societies and internal when it questions the individual's ability to think about the meaning of their actions, to use their own actions as a basis for analysing their origin and consequences (Akoun and Ansart 1999, pp. 441–42). Reflexivity is about engaging in the constant scrutiny of the self (Greene and Park 2021) and questioning your own ethical responsibility (Creswell and Poth 2016). Lapeyronnie writes that it conveys a form of self-analysis or "objectification of the self" (Lapeyronnie 2004, p. 639), which is a particular way of regarding and judging oneself in light of recent events that define one's existence. These current times, which have experienced massive upheaval because of the global COVID-19 pandemic (and also certainly due to global warming and the extinction of some species of animal), are particularly likely to prompt an individual to form an "awareness of themselves" (Touraine 2005, p. 157) by questioning the voluntary and involuntary, direct and indirect consequences of the social progress in which they play the role of both actor and driver. As we know, and as journalist Marie-Monique Robin (Robin and Morand 2021) has very clearly demonstrated from this perspective, successive pandemics are the consequence of our lifestyles and the changes that impact fauna and flora on a global scale. Reflecting upon the ethics of responsibility (Weber) is therefore essential. If we are to follow Giddens' and Habermas' thinking, this reflexivity represents an integral part of the modern age.

## 3. Reflexivity: The Hallmark of Modern Man

Giddens believes that the modern age is characterised by rationality and reflexivity. He writes that rationalisation is a long-term social process, in which traditional ideas and beliefs are replaced by methodical rules and procedures and formal reflection. Consequently, this rationality allows for habits and reflexes to be more easily called into question, "*rolling social life away from the fixities of tradition*" (Giddens 1994, p. 59). It questions ways of living and thinking. Acting rationally means acting reasonably. This demonstrates the capacity of the social individual to reflect on their actions and the consequences of these (while, in the Weberian tradition, the ethics of conviction bases action on values without regard for consequences, the ethics of responsibility views action solely in regard to the purposes and consequences of the actions). Modernity heightens reflexivity. As described by English sociologist, modernity is characterised by its high degree of reflex-

ivity. This author believes that reflexivity is a kind of continuous reflection by the social actor on themselves and their immediate social context. "*Social reflexivity refers to the fact that we have constantly to think about, or reflect upon, the circumstances in which we live our lives*" (Giddens 2006, p. 123). It is the process by which social practices are constantly examined and reformed in the light of incoming information about those very practices (Giddens 1994, p. 45). If this reflexivity is the hallmark of humanity, modernity gives it a more systematic and indispensable quality for shaping and maintaining social systems. What modernity gives humanity are the tools to express itself and think more freely. It should be noted that humans, according to Huizinga (1955), are Homo sapiens and Homo faber. They are beings of knowledge who seek to understand their environment. They progress by learning from their mistakes. However, they are also workers and creators, capable of transforming the world and creating their own environment by altering the natural space around them. More so today than in the past, they have the ability to think for themselves. They are aware that their actions can change the established way of things. "*Reflexivity is a defining characteristic of all human action*", writes Giddens (1994, p. 43). It is, he believes, a key characteristic of modernity. Although most authors agree that reflexivity is not a new concept or a skill discovered by humanity at a late stage, they nevertheless consider it to have been reinforced and augmented by modernity and the distancing of the influence of environmental constraints. It is, adds Kaufmann, "*the very opposite of an illusion, and its flourishing is even at the heart of the process of civilisation*" (Kaufmann 2001, p. 208). Particularly as, it should be noted, the Industrial Revolution and subsequent technological revolutions have greatly increased the time available for thinking. In 1800, 48% of a person's waking hours throughout the course of their life were spent working. In 2020, this percentage is just 11%. It is estimated that by 2025, 50% of work will be performed by robots. As regards available time for thinking, this has increased fivefold since 1900[6]. However, this reflexivity is not experienced equally by everyone. It presents varying levels of development and sophistication, depending on the individual in question.

## 4. Practical Consciousness and Discursive Consciousness

As highlighted by Molénat (2006, p. 51), modernity has certainly given individuals an understanding of their actions. They are reflective actors, i.e., they are capable of examining their actions, determining the conditions that gave rise to these actions and, to a certain extent, the consequences. However, these skills present themselves to varying degrees depending on the individual in question and their cognitive potential. In his two major works, published in 1987 and 1994, Giddens differentiates between two types of reflexivity, attesting to these varying degrees of human competence. Reflexivity, which forms part of "*practical consciousness*" (Giddens 1994, p. 92), refers to the minimal control that the individual is capable of exercising over their actions in the process of performing them. "*All human beings are competent agents*", writes Giddens. "*All social actors possess a remarkable understanding of the conditions and consequences of that which they do in their everyday lives*" (Giddens 1987, p. 343). This action relates to rules and procedures that form part of daily life. The wearing of masks, safety measures, and respect for social distancing rules illustrate this practical reflexivity that now features in the daily lives of the billions of human beings who inhabit the planet. The majority of people follow the health guidelines because they find them meaningful and useful, a way of protecting both themselves and others, as well as slowing the spread of coronavirus variants. Routinisation is the term used by Giddens to refer to all those daily tasks that become embedded in practical consciousness. Bourdieu, who does not subscribe to the idea that individuals possess inherent knowledge (Bourdieu 1993, p. 1413), also explains that they draw upon practical knowledge, or what he calls "*routines of ordinary thought*" (Bourdieu 1994, p. 9), on a daily basis, allowing them to access substantive realities, i.e., realities that are tangible and accepted as being self-evident (*Ibid.*)[7]. This pragmatic and causal understanding of the sequences of their actions allows individuals to influence their behaviour with relative efficiency (Dubet 1994, p. 225). However, there is a more sophisticated awareness that goes beyond practical awareness.

At a higher level, Giddens explains that reflexivity relates to "*discursive consciousness*". This consciousness is exercised when the individual manages to explain the reasons for their action in a coherent manner, when they can "*put things into words*" (Giddens 1987, p. 93). In view of the fact that this higher form of reflexivity is essentially accessible through interactive discourse, it is important to stimulate debate and fuel conversation in order to prompt individuals to develop their ability to situate themselves in relation to both themselves and others by means of discussion and exchange. However, as Giddens (1994, p. 60) reminds us, this knowledge remains a sociocultural marker because it is often only accessible, in its varying degrees, to individuals who possess the power and cognitive skills (this differing capacity explains why some people are more influential than others, some nations more advanced than others). Discursive reflexivity is not available to everyone.

### 5. The Dual Reflexivity of Knowledge and the Role of Whistle-Blowers

This reflexivity is that found in rational individuals, but it cannot be fully expressed without the actions of scientists, intellectuals, influencers, and whistle-blowers. This is the "*dual reflexivity of knowledge*" described by Vrancken (2001, p. 319). Doctors Li Wenliang and Ai Fen were the first to issue a warning about the danger presented by the COVID-19 coronavirus. At the end of December 2019, the former alerted the population in a discussion forum before succumbing to the virus a few weeks later. The latter spoke to *Renwu* (People) magazine to reach the general public. Other scientists, bloggers, and journalists, such as the Russian journalist Tatiana Baïs and the Chinese journalist Chen Qiushi, then relayed the Chinese doctors' messages. There had also been warnings long before the COVID-19 public health crisis. SARS respiratory syndrome, Ebola virus, Lassa fever, HIV/AIDS, H1N1 avian flu, and chikungunya virus are all "zoonotic" diseases, i.e., they are transmitted from animals to humans as a result of profound transformations within ecosystems (transformations brought about by uncontrolled urbanisation and policies permitting unchecked deforestation). All these viruses have been the subject of publications warning of their potential danger by scientists such as Kate Jones, Thomas Lovejoy, Karl Johnson, Malik Peiris, James Mills, and Serge Morand. This is discussed by journalist Marie-Monique Robin in a book published by Robin and Morand (2021), which explains that the destruction of biodiversity is likely to be the main factor in the emergence of infectious diseases. After the period of shock that preceded the lockdown and closure of schools, public awareness became real. This was accompanied by educational and explanatory actions by scientists and political leaders. An online survey conducted in India to examine the general public's knowledge, attitude, and anxiety levels during the COVID-19 pandemic showed that 98% of participants acknowledged the need for social distancing (Anderson et al. 2020). A similar level of awareness among the general public has also been observed in other countries, such as Nepal (Hussain et al. 2020) and Kenya (Austrian et al. 2020). According to a 2020 study carried out in Saudi Arabia (Alahdal et al. 2020), 80% of those surveyed stated that they had changed their behaviour due to the virus. While these studies show that the majority of the population accepted the healthcare and preventive measures, other surveys (Jun et al. 2021) confirm that the pandemic raised the public's awareness regarding shared responsibilities, resulting in an increase of more than 20% in the number of searches for information on the coronavirus, most notably via Google searches. In another study (Alam et al. 2021), 70% of participants reported being knowledgeable about the new disease. The most common sources for acquiring information are social media (Abuhashesh et al. 2021), followed by radio and television (Owhonda et al. 2021). Although this awareness is admittedly sudden, explains Le Monde[8], it helps to open up a debate that has often remained limited prior to now. The evolution of the pandemic legitimises information gathering by the population and therefore contributes to an urgent growth in individual and collective awareness.

**6. Bringing about the Cognitive Conditions for Self-Reflexive Analysis**

The individual certainly assumes greater responsibility on the whole today than in the past, but this is the consequence of an increased awareness of their reflexive power and ability to understand the world around them. Endowed with the skills of reflexivity, people most commonly mobilise their practical consciousness and regularly mobilise their discursive consciousness to explain how they act and why they act this way. In view of the fact that this higher form of reflexivity is principally accessible through interactive discourse, it is important that debate is initiated and that conversation is nurtured in order to prompt individuals to develop their ability to become oriented in relation to themselves, others, and the world in which they live. This is similar to Bourdieu's method, with the countless interviews he conducted with people who were suffering. It is a matter of creating the conditions for a "prompted and supported self-analysis" (Bourdieu 1993, p. 1408). When questioned about their lives, people are encouraged to recount their daily lives and question themselves. By engaging in "*continuous commentary on their activities*" (Molénat 2006, p. 53), they thereby undertake useful, even life-saving, explanatory work. When faced with threats and dangers, individuals become aware of their individual and collective responsibility, and of the actions that they can take to change things. Reflective work is a dynamic process of self-construction, says Giddens. Simply having an attentive interlocutor prompts individuals to express themselves and find the words to expose their doubts and feelings of unease. To a certain extent, the whistle-blower helps to create favourable conditions for this self-reflexivity. Of course, not all influencers are a force for good. As Bronner explains, the cognitive bubble also shows forms of echo chamber thinking; closed, community-based worlds that can fuel fake news and misinterpretations. The sought-after "proxemic" (a term used by Hall 1974) aims to reduce the information gap between individuals to the greatest possible extent. Individuals use social media in particular to inform each other about political, social, and cultural events without always relying on journalists, scientists, and experts on the topics addressed, who are sometimes considered untrustworthy (Massé et al. 2011). Consequently, there is a risk that untruths and other conspiracy theories will be perpetuated. However, it is then up to the whistle-blowers and influencers, whether they be scientists, intellectuals, or ordinary citizens, to open up the debate and create the conditions to affirm, on a basis of reason and rationality, the dual reflexivity of knowledge and the self-analysis of the thinking subject.

**7. Awareness in a State of Crisis**

Giddens, of course, knows that people cannot explain everything verbally, nor examine all their actions in a lucid and cognitive light, many of which are unintentional. Some forms of cognition are repressed and can be considered as an unconscious state. Some intentions do not lead to action and some actions are not carried out intentionally. Expanding upon the English sociologist's thinking, researchers such as Dubet, Kaufmann, Lahire, and Hamel (2007) have questioned the role of reflexivity in explaining human behaviour. All these debates question the ambivalent nature of the actor's reflexivity, which alternates between permanence and exception, oneness and fragmentation, novelty and continuity. It is the view of some authors that reflexivity does not appear spontaneously. Bourdieu stated that we are reflexive in a state of crisis, which is when there is an imperative to adapt to a new or unusual situation, when there is a pressing obligation to change our habits and customs. Kaufmann (2001) concentrates on the fragmented and multifaceted dimension of reflexivity, which he regards as uncertain and lacking in unity. For Lahire (2001), reflexivity is by no means unified. Humanity uses a multitude of minor reflexivities that arise in daily life on an ongoing basis, in moving from one social sphere to another, when they are confronted with cultural (moving house and population displacement), social (accident and isolation), or even individual (dismissal and death) disturbances. Putting Giddens' hyper-reflexivity into perspective, Ehrenberg does not believe that lifestyles are chosen freely. He adds that it is wrong to believe that "*the individual alone, subjectively and by virtue of their reflexive capacity, generates the social link in their interactions with other*

*subjects*" (Ehrenberg 2005, p. 201). Contemporary individualism is in fact the result of an extended maturing period of the human species, born of progress and learning, but also of forgetfulness, repression, and resilience. This individual and collective reflexivity must be sought out, brought about, and maintained through warning signs that pre-empt disasters, by means of scientific publications, messages in the national and local press, and clips and videos posted on social networks, which, once the state of shock has passed, will allow for reflection and a profound and lasting change in human activities.

## 8. Conclusions: Reflexivity of Popular Knowledge

The successive lockdowns, the alarming development in the numbers of daily deaths, and the huge coronavirus vaccination campaigns are all circumstances that trigger what Massé, Weinstock, Désy, and Moisan refer to as a "reflexivity of popular knowledge" (Massé et al. 2011). Faced with a threat that is indeed on their doorstep, humans develop a capacity to adapt, to question, and to adopt a critical perspective (admittedly varying in its degree of permanence) that is likely to alter behavioural habits in the long term. Awareness can be life-saving if it is followed up with actions. In the case of the pandemic, this involves changing lifestyles and consumption habits, preserving natural areas and biodiversity, restoring ecosystems, stopping invasive species, combatting climate change, and curbing energy consumption. The road is long and full of obstacles, and there are other disasters yet to come, but the efforts of individual and collective reflexivity are essential in pointing the way to a safer path, via which we can protect the planet and all the living beings that inhabit it.

**Funding:** This research received no external funding.

**Data Availability Statement:** Not applicable.

**Conflicts of Interest:** The authors declare no conflict of interest.

## Notes

[1]  American College Health Association, 2015. National College Health Assessment II: Reference Group Executive Summary Spring 2015. Hanover, MD: ACHA.

[2]  This survey was conducted online within the United States by The Harris Poll on behalf of the American Psychological Association between 11 August and 23 August 2021 among 3035 adults age 18+ who reside in the United States.

[3]  Lancet Global prevalence and burden of depressive and anxiety disorders in 204 countries and territories in 2020 due to the COVID-19 pandemic.

[4]  This 2021 study was carried out by Dynamic Workplace and Speak & Act, in partnership with Herman Miller, the Île-de-France region, Hôpital Européen de Paris, and INSEEC's MSc & MBA and Bachelor programmes.

[5]  Interview with Jürgen Habermas for the newspaper Le Monde, "During this crisis, we must act in the explicit knowledge of our lack of knowledge", published on 10 April 2010 (interviewed by Nicolas Truong).

[6]  Figures reported by Bronner in his book published in 2021 (cf. in Bronner 2021, particular pp. 78 to 86).

[7]  It should be noted that, according to Bourdieu, the reflexivity of the individual stops at this practical and routine level. Due to the difficulty involved in distancing themselves from common sense, the author believes that individuals cannot, for the most part, comprehend the principle underlying their action, unease or dissatisfaction (Bourdieu 1993, p. 1413). By contributing theoretical knowledge to the practical knowledge of the individual (Bourdieu talks about "*knowledge of knowledge*"), the sociologist thus fully fulfils their explanatory role within society.

[8]  Editorial in the newspaper Le Monde dated 17 March 2020 entitled "*Coronavirus: le pari de la prise de conscience citoyenne*" ("Coronavirus: the gamble of raising public awareness").

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
