# Peer review of "Health Crisis and the Dual Reflexivity of Knowledge"

_socsci, doi:10.3390/socsci11040161_

Round 1
Reviewer 1 Report
The manuscript titled: “Health Crisis and the Dual Reflexivity of Knowledge” is a discussion paper. In my own opinion, the paper attempts to argue from a reflexivity conceptual framework that pandemic or any health crisis, apart from their adverse effects, do have some noticeable positive effects in society. One general question that comes to mind immediately is to ask the authors why they think the act of reflexivity is not experienced outside (the boundary of) health crises or pandemics? See lines 59- 66!
In addition, I would like to point at the following cohesive and structural issues to the authors:
- In lines 4 and 50, do you think the word “more” is appropriate here? Because it may also connote that pandemics or health crises create more positive effects than adverse effects!
- In lines 8 and 55, the word “this” may also connotes that Habermas use of the concept of reflexivity is confined to the boundaries of pandemics and health crises. If it is so, why was Habermas not explicitly quoted or summarized to make this point explicit. In particular, Habermas, Weber, and Giddens constitute the major points of view on the concept of reflexivity in the paper. Why did authors discuss Weber’s and Gidden’s in Section 3 and left out Habermas?
- Why did the authors not explicitly state the aim of the paper? Why are Sections 1-5 limited to one long paragraph each? Why is the last paragraph in Section 6 not labeled as the conclusion section?
In my own opinion, if the authors could address the issues above, their noble discussion of the “…Dual Reflexivity of Knowledge” would be easier to follow by the intended audience.
Author Response
We have given a broader definition of reflexivity by integrating the external dimension of reflexivity turned towards the social conditions of knowledge production in modern societies. Lines 4 and 50, we have deleted the inappropriate and connotative word "more". In lines 8 and 55, we have replaced the word 'this' with the word 'the' in order to open the discussion to Habermas' conceptions and his more historical and political definition of the concept of reflexivity. We have clarified, in the abstract and introduction, the purpose of our paper. We have also better identified our concluding paragraph. We hope that we have answered the criticisms of reviewer.
Thank you very much for your confidence.
Reviewer 2 Report
The paper offers an interesting discussion on the notion of “reflexivity” that becomes very important in this pandemic period. It refers to relevant sociological and philosophical studies on Modernity by presenting a plausible view on human possibilities of self-reflection even though our nature is also a “second nature” expressed by habits and social routines. I appreciate the positive analysis of the notion of habit that considers the possibility of an active dimension to be exercised in discursive practices. I think that the paper is clear and well discussed so I suggest to publish it.
Author Response
Dear colleague,
Your criticism of our text is good and we thank you very much for your confidence.